

# A Validation and Code-to-Code Verification of FAST for a Megawatt-Scale Wind Turbine with Aeroelastically Tailored Blades

Srinivas Guntur*[1], Jason Jonkman[1], Ryan Sievers[2], Michael A. Sprague[1], Scott Schreck[1], and Qi Wang*[1]

[1]National Renewable Energy Laboratory, Golden, CO, USA
[2]Siemens Wind Power, Inc., Boulder, CO, USA

*Correspondence to:* srinivaskguntur@gmail.com

**Abstract.** This paper presents validation and code-to-code verification of the latest version of the U.S. Department of Energy, National Renewable Energy Laboratory wind turbine aeroelastic engineering simulation tool, FAST v8. A set of 1,141 test cases, for which experimental data from a Siemens 2.3 MW machine have been made available and were in accordance with

5 the International Electrotechnical Commission 61400-13 guidelines, were identified. These conditions were simulated using FAST as well as the Siemens in-house aeroelastic code, BHawC. This paper presents a detailed analysis comparing results from FAST with those from BHawC as well as experimental measurements, using statistics including the means and the standard deviations along with the power spectral densities of select turbine parameters and loads. Results indicate a good agreement among the predictions using FAST, BHawC, and experimental measurements. These agreements are discussed in detail in this

paper, along with some comments regarding the differences seen in these comparisons relative to the inherent uncertainties in such a model-based analysis.

## 1 Introduction

It is well known that the wind energy industry is growing rapidly worldwide, and increasingly larger blades are being employed by state-of-the art wind turbine technology. With the advent of large, flexible blades, the tools required to design and analyze

these turbines also need to be updated according to the current and future state of affairs. A wind turbine aeroelastic simulation code consists of several modules that model different aspects of wind turbine physics (aerodynamics, structures, pitch/torque control modules) (Hansen, 2000; Zhang and Huang, 2011). Each of these modules is based on a set of assumptions that may or may not be valid as the size and flexibility of the turbine increases. For example, torsional degree of freedom (DOF) becomes increasingly important for larger, more flexible blades, especially those that use sweep-twist or bend-twist coupling for passive

load alleviation (Rasmussen et al., 2003; Zhang and Huang, 2011). The next generation of aeroelastic codes should be capable of modeling such structures. The latest version of FAST incorporates some of these advancements, as explained later in the paper.

Although one challenge is the availability of models that can capture physics relevant to state-of-the-art wind turbines, the other and an equally important challenge is validation of such codes. Historically, there have been several international

---

*Currently at Siemens Wind Power, Inc.





industrial as well as academic efforts to carry out code-to-code verification and validation of wind turbine models against experiments; most notable among others are the MEXICO (Schepers et al., 2012, 2014) and the National Renewable Energy Laboratory (NREL) UAE Phase VI (Simms et al., 2001) blind test comparisons. These experiments offered the response of scaled, experimental wind turbines in controlled conditions in wind tunnels. Although these test campaigns have provided

high-quality data capturing various aerodynamic effects, structurally these turbines were stiff.

Published works on validation of high-fidelity aeroelastic codes using larger, flexible turbines, such as Kallesoe and Kragh (2016), are rare mainly due to the high costs and physical challenges associated with such an endeavor. To the authors' knowledge, full system validation of wind turbine aeroelastic codes for a wide range of realistic operating conditions, as recommended in Guzman and Cheng (2016), for example, are extremely rare to find in published literature despite their relevance

to the state of the art in wind turbine research and development. The current work, which has been the result of a research collaboration between NREL and Siemens Energy over a period of 3 years, presents the full system validation of FAST, using detailed and high-quality experimental measurements on a 2.3 MW machine with flexible, aeroelastically tailored blades.

## 1.1 FAST v8

FAST is an open-source multiphysics simulation tool that was created for design and analysis of advanced land-based and

offshore wind technology. Underpinning FAST is a modularization framework that enables coupling of various modules, each representing different physics domains of the wind system (Jonkman, 2013; Sprague et al., 2015). FAST version 8 (FAST v8) includes a mesh-mapping utility allowing each module to be independently discretized in space and time, and a mathematically rigorous solution procedure supporting loose coupling of modules with implicit-coupling relations. For land-based turbine simulations, FAST has modules for wind inflow (InflowWind); aerodynamics (AeroDyn); control and electrical-drive dynamics

(ServoDyn); and blade, drivetrain, nacelle, tower, and platform structural dynamics (ElastoDyn).

In the newest release of FAST, BeamDyn (Wang et al., 2015), a new blade-dynamics module, has been introduced and the AeroDyn aerodynamics module has been overhauled (Ning et al., 2015) to support the analysis of advanced aeroelastically tailored blades. BeamDyn is based on geometrically exact beam theory and is implemented using Legendre spectral finite elements. The geometric nonlinearities, including large displacements and rotations in a three-dimensional space, are captured

by BeamDyn without introducing ad-hoc assumptions and have initially curved and twisted reference lines. Specifically, it supports built-in curve, sweep, and sectional offsets of wind turbine blades. Moreover, along with a cross-sectional analysis tool like VABS (Wang and Yu, 2011) and BECAS (Blasques et al., 2016), BeamDyn is capable of simulating the elastic deformations (extension, shear, bending, and torsion) and the coupling effects between all six DOFs for both isotropic and composite slender structures. When these advanced features are needed, BeamDyn replaces the more simplified blade structural

model of ElastoDyn, which is only applicable to straight isotropic blades dominated by bending.

Finally, in the latest FAST v8, AeroDyn has been overhauled to (1) fix underlying problems with the original theoretical treatments, (2) introduce improved skewed-wake and unsteady aerodynamics models, (3) enable modeling of highly flexible and nonstraight blades, and (4) support the unique features of the FAST modularization framework (Ning et al., 2015; Damiani et al., 2016).





## 1.2 Experimental data

The National Wind Technology Center (NWTC) at NREL is home to several megawatt-scale test turbines. One of these is a Siemens 2.3 MW machine (SWT-2.3-108) that operates in an upwind configuration and is equipped with a three-bladed, 108 m diameter rotor (Medina et al., 2012). It is a variable-speed turbine operating under a collective pitch control, and the rotor

is connected to the generator via a gearbox. The blades used on this machine have a prebent geometry (the blade shape under no-load conditions is not straight, but curved). Additionally, these blades are flexible, aeroelastically tailored, and incorporate bend-twist coupling. This turbine is instrumented with: Fiber-Bragg strain sensors capable of measuring blade-tip deflections; surface pressure taps; strain gages at the blade root, tower top, and the tower base; and a data-acquisition system that records turbine operating data such as the rotor speed and electrical power. Wind speed and direction are measured by sensors mounted

on an upwind meteorological tower (met tower). Measurements on this heavily instrumented turbine have been collected and shared as part of a cooperative research project between NREL and Siemens Energy. This unique set of measurements is well suited for validation of high-fidelity blade structural modeling tools, like BeamDyn, and facilitates a wide variety of research possibilities. In the current work, the comparisons between FAST and BHawC predictions and the measurements will be discussed for various turbine operating conditions consistent with the recommendations in the International Electrotechnical

Commission (IEC) 61400-13 standard (IEC 61400-13, 2001).

## 1.3 Current work

This paper presents a detailed validation and code-to-code verification of the latest version of the U.S. Department of Energy's open-source wind turbine aeroelastic simulation tool FAST v8, which is supported by NREL. A previous paper (Guntur et al., 2016) presented preliminary results from comparisons among predictions from FAST, the Siemens in-house aeroelastic code

BHawC, and the experimental data from the Siemens 2.3 MW wind turbine. Following the preliminary analysis presented in the previous study, FAST has undergone some improvements, which are highlighted in this paper. This paper presents a validation and code-to-code verification of FAST v8 using results from the improved FAST model analyzed in time as well as frequency domains.

## 2 Methods

The analysis presented in this paper is centered on the Siemens 2.3 MW turbine with a 108 m rotor that is installed at NREL's NWTC. The turbine is heavily instrumented, and measurement data have been made available through a collaborative research between NREL and Siemens Energy. Data were collected over a period of several months under normal operating conditions for a range of wind speeds and turbulence intensities, and these data were used in this validation exercise. Validation was conducted following the guidelines stipulated in the IEC wind turbine loads-measurement standard (IEC 61400-13, 2001).

Code-to-code verification was accomplished by comparing FAST v8 simulation results with those of the Siemens in-house aeroelastic simulation tool, BHawC.



## 2.1 Experimental measurements

A large amount of test data was recorded on the Siemens 2.3 MW wind turbine and the NWTC 135 m met tower. The met
tower is located approximately 2.5 rotor diameters upstream of the turbine and is instrumented with several sensors along its
length that measure the wind speed, wind direction, and atmospheric pressure. The inflow wind speed data used in this analysis

were recorded at 80 m, which is close to the turbine hub height. Each 10 min data set contains surface-pressure measurement
data, strain-gage data, turbine-operation data, and the inflow data from the met tower. In total, several months of data have been
recorded, of which this report presents data from 1,141 10 min data sets. The data samples were selected based on their mean
hub-height wind speed and turbulence intensities in order to populate the recommended test matrix of the IEC 61400-13 (see
Table 1). While the 1,141 simulations were run according to Table 1, the results plotted in Section 3 are binned only by wind

speed. Overall, these data represent operating conditions with inflow velocities between the turbine's cut-in and the cut-out
wind speeds, at various inflow turbulence intensity levels, up to approximately 23%.

| Turbulence | Mean Wind Speed [m/s] | | | | | | | | | | | | | | |
|---|---|---|---|---|---|---|---|---|---|---|---|---|---|---|---|
| | Region 2 | | | | | | | | v (rated) | Region 3 | | | | | |
| Intensity (%) | 3.5-4.5 | 4.5-5.5 | 5.5-6.5 | 6.5-7.5 | 7.5-8.5 | 8.5-9.5 | 9.5-10.5 | 10.5-11.5 | 11.5-12.5 | 12.5-14 | 14-16 | 16-18 | 18-20 | 20-22 | 22-24 |
| < 3 | 0 | 4 | 6 | 3 | 0 | 0 | 0 | 0 | 0 | 0 | 0 | 0 | 0 | 0 | 0 |
| 3-5 | 0 | 20 | 19 | 19 | 9 | 5 | 1 | 1 | 0 | 0 | 0 | 0 | 0 | 0 | 0 |
| 5-7 | 0 | 32 | 50 | 32 | 14 | 3 | 7 | 3 | 2 | 2 | 0 | 0 | 0 | 0 | 0 |
| 7-9 | 0 | 43 | 39 | 44 | 17 | 17 | 3 | 7 | 2 | 2 | 2 | 0 | 1 | 0 | 1 |
| 9-11 | 0 | 32 | 42 | 35 | 29 | 13 | 7 | 11 | 9 | 5 | 9 | 5 | 0 | 0 | 1 |
| 11-13 | 0 | 22 | 21 | 16 | 16 | 9 | 12 | 15 | 16 | 14 | 15 | 6 | 4 | 1 | 0 |
| 13-15 | 0 | 2 | 16 | 25 | 10 | 6 | 14 | 15 | 24 | 11 | 15 | 14 | 5 | 1 | 0 |
| 15-17 | 0 | 1 | 10 | 11 | 6 | 4 | 7 | 12 | 12 | 10 | 18 | 8 | 9 | 2 | 0 |
| 17-19 | 0 | 2 | 12 | 6 | 0 | 2 | 3 | 12 | 4 | 10 | 8 | 7 | 5 | 0 | 0 |
| 19-21 | 0 | 2 | 3 | 3 | 2 | 1 | 3 | 1 | 4 | 2 | 3 | 1 | 0 | 0 | 0 |
| 21-23 | 0 | 0 | 0 | 1 | 0 | 0 | 0 | 0 | 0 | 1 | 0 | 0 | 0 | 0 | 0 |
| Sum | 0 | 160 | 218 | 195 | 103 | 60 | 57 | 77 | 73 | 57 | 70 | 41 | 24 | 4 | 2 |

**Table 1.** IEC matrix populated with data bins defined by mean wind speed and turbulence intensity. The numbers in the table represent the number of 10 min data sets within each bin, for a total of 1,141.

Each bin in the test matrix (Table 1) contains several 10 min data sets. For each data set, the mean, standard deviation, and
the power spectral density (PSD) of the following quantities of interest (QOI) are determined from both the simulations and
the measured data:

– Rotor speed

– Electrical power

– Blade-root bending moments (in and out of the rotor plane)

– Main-shaft bending moments (yaw and tilt directions, in a nonrotating coordinate system)

– Tower-top torsional moment

– Tower-bottom bending moments (parallel and perpendicular to the mean wind direction)



- Blade-tip deflections (in and out of the rotor plane).

For simplicity, only select PSDs are presented in this paper at three different operating conditions for each QOI: one at below-rated, one at rated, and one at above-rated hub-height inflow wind speed.

## 2.2 Aeroelastic model of the SWT-2.3-108 machine

The FAST model of the SWT-2.3-108 machine was created based on the information obtained from Siemens as well as the inflow data from the experimental database for each of the cases simulated. This three-bladed upwind turbine has a fixed coning angle of $5°$ and a rotor tilt angle of $6°$, both directing the blades away from the tower. The tower, blades, and the main shaft are modeled as elastic bodies with properties obtained from Siemens, and the tower base is fixed. Data used for model calibration are generally recommended to be different from that used for validation (see e.g., Bayarri et al. (2007)), and accordingly, the
information used to build the FAST model for the purpose of validation in this study has not been used for model calibration.

Details of the FAST v8 model have been described in a previous paper (Guntur et al., 2016). A part of this is reiterated below for thoroughness in the model description, and also to provide the reader with the relative improvements since the last version of the turbine model.

In the BeamDyn, it is possible to model a blade defined by many cross-sectional property stations with relatively few node
points for integration while capturing all of the provided material properties (Wang et al., 2016). The cross-sectional properties of the blades on the Siemens machine were defined at 106 stations along its length. For BeamDyn spectral finite element calculations (e.g., those for mass and stiffness matrices), these data were spatially integrated using the trapezoidal rule, and each blade was discretized using a single seventh-order spectral finite element. In contrast to the nonlinear finite-element implementation of BeamDyn, the structural model of BHawC employs a co-rotational beam formulation, which is a combined
multibody and linear finite-element representation allowing for geometric nonlinearities through a series of multiple bodies, each composed of a linear finite element. The BHawC model of the SWT-2.3-108 blade used in the current study was initially curved in space and discretized into 16 linear elements.

The two-dimensional airfoil aerodynamic properties for the blade were obtained from Siemens. The blade was modeled in AeroDyn using 20 nodes. The lift-coefficient data were processed to include rotational augmentation effects in the same way
as BHawC, according to a variation of the Snel et al. (1993) model.

The measured yaw error in most test cases was very small, with a few exceptions of yaw errors higher than $5°$ (based on a 10 min average of the wind-direction sensor readings obtained from the upwind met tower). Both BHawC and FAST simulations take this into account. In the FAST simulations, a fixed nacelle yaw error was used with a value equal to the 10 min average yaw error for each case.

FAST used the same turbulent inflow input files as BHawC, which used HawC-style turbulence boxes. For each case simulated, the turbulence box is scaled according to the mean wind speed and the turbulence intensity at the turbine hub height, obtained from the 80 m wind speed and direction measurement from the met tower.

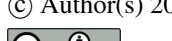



A controller in a Bladed-style Dynamic-Link Library (DLL) form obtained from Siemens was used for pitch and torque control in the FAST simulations. It is known that this DLL controller does not employ the exact same control logic as the physical turbine or the BHawC simulations, but is similar.

### 2.2.1 Improvements to the FAST model of SWT-2.3-108

The following are improvements to the current FAST model over the model used in the previous study (Guntur et al., 2016):

- In the FAST simulations in this paper, wind shear was modeled assuming a power-law profile. In the previous paper, a power-law exponent of 0.2 was assumed for all 1,141 simulations, whereas in this work, the shear exponent was derived using experimental measurements. The met tower collected wind speed and direction information at the heights of 3 m, 10 m, 26 m, 80 m, 88 m, and 134 m. These data were used to estimate a power-law exponent by the least-squares
best-fit method. For the few cases where a best-fit approximation was deemed unsuitable (e.g., low-level jet boundary layer profile), a power-law coefficient of 0.2 was assumed. The BHawC simulations used atmospheric shear estimates based on lidar measurements, and so there may be small differences among the inflow conditions of the test turbine, FAST simulations, and the BHawC simulations, adding to the uncertainty in this exercise.

- It was found that the rotor mass imbalance was modeled too high in the previous paper and has been corrected in the
15 current model.

- A part of the experimental measurements of the main-shaft bending moments was found to contain corrupt data, and those data have been filtered out.

- A controller in a Bladed-style DLL form obtained from Siemens was used for pitch and torque control in the FAST simulations. In the previous study, it was observed that the FAST simulations with the Siemens DLL controller gave
rise to some resonance effects in the coupled drivetrain and tower side-side modes, which in the previous study were suppressed by artificially increasing the drivetrain damping. It was later found that the FAST-DLL controller interface requires a low-pass filter to filter out high-frequency content on the generator speed in order for the FAST model to represent the control mechanism as close to the BHawC model/experimental turbine as possible. This filter was added to FAST in the simulations shown in this paper, resolving the resonance issue highlighted earlier. The drivetrain properties
used in the current simulations are those that were prescribed by Siemens (i.e., the damping was reduced to its original value in the new results); no further artificial changes have been made.

- The previous study did not include unsteady aerodynamics effects, which are included in this work (Ning et al., 2015; Damiani et al., 2016).

The simulations shown in this work were generated using FAST v8.15.01a-bjj compiled in double precision with:

- ElastoDyn v1.03.02a-bjj,



- BeamDyn v1.01.01,

- AeroDyn v15.00.01a-bjj,

- InflowWind v3.02.00a-adp, and

- ServoDyn v1.04.00a-bjj.

The FAST simulations with BeamDyn used a time increment of 0.0005 s, whereas those with ElastoDyn used 0.005 s. All FAST simulations were carried out for 12 minutes, where the first 2 minutes were ignored to remove initial-transient effects. BHawC simulations were 11 minutes long with a 0.02 s integration time increment, and the first minute of transience was ignored. Note that the structural blade in ElastoDyn is straight and only includes bending DOFs, whereas the structural blade in BeamDyn is curved (as is the blade model in BHawC) and includes the DOFs bending, torsion, shear, and extension, with composite coupling terms.

## 3 Results

The plots presented herein show the mean and the standard deviation values of several QOIs for each of the 1,141 10 min sample cases simulated, along with the PSDs of select bins at below-rated, at-rated, and above-rated operating conditions. The $y$-axes of the time-series plots as well as the $x$-axes of the frequency spectra plots have been removed to protect Siemens proprietary data, and so this paper presents a discussion based only on the relative trends among the different data shown. Each figure shows four different comparisons of the same QOI.

In the previous paper, each 10 min test case was simulated using three different FAST configurations, to be able to analyze the relative improvements in FAST among the capabilities in its previous public releases (FAST v8.10 and earlier) and the current developments with BeamDyn and AeroDyn. One observation from the previous study has been that the effect of the aerodynamic blade curvature on the FAST results is negligible. Therefore, this study presents comparisons of FAST simulations with straight ElastoDyn blades and with curved BeamDyn blades, where both used curved aerodynamic blades. In the figures that follow, the different data identified in the legends are as follows:

- *Data* - experimental measurements

- *BHawC* - results from Siemens' BHawC simulations

- *FAST (ED)* - FAST simulations with ElastoDyn

- *FAST (BD)* - FAST simulations with the new BeamDyn.

Note that "ElastoDyn" and "BeamDyn" designate structural modules used to model the blades only; the structural dynamics of the remainder of the turbine (drivetrain, nacelle, and tower) were always modeled in ElastoDyn.

For Figures 1 to 11, subfigures (a) and (b) show a comparison of the data from FAST with BeamDyn, BHawC, and experimental results, whereas subfigures (c) and (d) show those comparing FAST with BeamDyn with FAST with ElastoDyn.



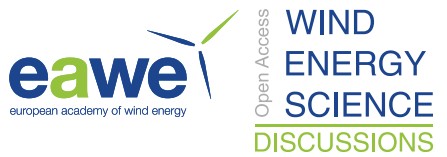

Subfigures (a) and (c) show the means of the individual 10 min data, while the subfigures (b) and (d) show the binned average means of the same QOIs, and the standard deviations calculated from the individual 10 min data in a given bin. The data points connected by a line indicate the mean value of the binned data, calculated as the average of the means of the individual 10 min simulations within that bin. The scatter above and below each of the mean data shows one standard deviation of the individual

10 min data sets in that bin added and subtracted to the mean of that bin.

Figures 12 to 22 show the PSDs of three select bins to represent three hub-height wind speed bins: 8 m s$^{-1}$, 12 m s$^{-1}$, and 15 m s$^{-1}$ (i.e., below-rated, rated, and above-rated wind speeds). Subfigures (a), (c), and (e) show comparisons of FAST with BeamDyn, BHawC, and experimental data, and subfigures (b), (d), and (f) show those of FAST with BeamDyn and FAST with ElastoDyn.

**3.1   Discussion of the results**

The current results consist of analyses in the time and frequency domains. The results indicate that the predictions from the latest version of FAST compare consistently well with those of BHawC as well as the experimental data. A comparison between the results from various FAST simulations indicate an improvement of the FAST simulations using BeamDyn over those using ElastoDyn.

The BHawC data shown in this paper are the same as what were shown in Guntur et al. (2016). The experimental data are also the same for all channels except the main-shaft bending moments, which have been filtered for some corrupt data that were identified. The FAST results shown here are generated using the FAST model with the new improvements discussed earlier in this paper.

The agreement among the results from the improved FAST model, BHawC, and experimental measurements remains very

good. Below is a discussion regarding the observations from the comparisons presented in the figures, and improvements to the results published in the previous study as a result of the newer FAST model.

**3.1.1   Comparisons using the means and the STDs**

Figures 1 and 2 show the electrical power and the rotor speed, respectively. Note that each point in the scatter shown in subfigures (b) and (d) represents the standard deviation of the time series of an individual 10 min data set, added and subtracted

to the mean of that bin. The agreement between the mean values of different data in electrical power is excellent. Figures 1a and 2a also show that there are some isolated cases in region 2 where the rotor speed and electrical power deviate from the general trend for this turbine, and it is worthwhile to note that these data have some influence on the variance in the loads data presented next. These deviations are most likely due to a difference in the actual inflow field seen by the turbine compared to modeled turbulent inflow used in the aeroelastic models. Even so, the agreement in electrical power and the rotor speed

between FAST, BHawC, and the experimental data is close enough to assure that a majority of the turbine operating conditions being simulated and compared across FAST, BHawC, and the measurements are similar.

Figure 3 shows comparisons in the rotor thrust force. The rotor thrust is a difficult quantity to measure directly, and therefore, the measured thrust shown in this figure was estimated as the tower-bottom bending moment in the fore-aft direction divided by





the tower height. Examining Figures 3b and 3d, it is evident that the BeamDyn model for the blade provides a better estimate of the thrust compared to ElastoDyn, particularly around rated conditions. At higher wind speeds, FAST simulations appear to show higher standard deviation values in the rotor thrust compared to those from measurements and BHawC results. On the other hand, the agreement between FAST and BHawC in the mean looks excellent, and the agreement between FAST and the

experimental data looks reasonable.

Figures 4 and 5 show the in-plane and out-of-plane blade-tip deflections from the simulations and the experimental measurements. The blade-tip deflection was not measured directly but was estimated using the FiberBragg strain measurements from several locations along the span of Blade B. The local curvature was estimated from the strain measurements, which were then used to calculate the instantaneous shape of the blade. By integrating the shape, the magnitudes of the blade-tip deflections, in

and out of the rotor plane, were obtained. From Figures 4c and 5c, a significant difference between BeamDyn and ElastoDyn is visible. This difference is in fact an improvement, as revealed by Figures 4a and 5a. The agreement of FAST with BeamDyn with the measurements as well as BHawC is very good throughout regions 2 and 3, and the agreement with the experimental data is particularly good in region 3. These figures show that BeamDyn is a significant improvement over ElastoDyn. Note that only a fraction of the total set of experiments included reliable FiberBragg data−for example, the 8 m s$^{-1}$ bin contained 103 10

15    min data sets according to Table 1, but only 21 of these contained usable FiberBragg data. Because the PSDs of the blade-tip deflections shown in these figures were computed using fewer data sets, more noise is visible in these plots compared to other QOIs.

Figures 6 and 7 show the in-plane and out-of-plane blade-root bending moments, respectively. Experimental data for the blade-root bending moments consist of strain-gage measurements at the roots of all three blades. For simplicity, only data from

one of the three blades (Blade A) are being shown for the blade-root bending moments in this study. From the comparisons shown in Figures 6a and 6b, fairly good agreement can be seen among the BHawC, experimental, and FAST (with BeamDyn) data. The in-plane blade-root bending moment is heavily sinusoidal (the amplitude of the oscillations is much higher than the mean, which is the reason why the standard deviations shown in Figures 6b and 6d are so far away from the mean). From the out-of-plane blade-root bending moments shown in Figures 7b and 7d, the overall agreement among FAST with

BeamDyn, BHawC, and the measurements appears to be good. A small difference between the FAST results and the Siemens data is discernible, which may most likely be related to the small difference in the rotor speed also seen between FAST and Siemens data in Figure 2, as discussed earlier. Furthermore, from Figures 7c and 7d the improvement of FAST simulations with BeamDyn over ElastoDyn is clearly visible, more so at higher wind speeds.

The main-shaft yaw and tilt bending moments shown in Figures 8 and 9 are strong functions of the drivetrain properties. As

noted before, two improvements have been made to the experimental data shown in these figures: the experimental data have been processed to filter out corrupt data, and modifications to the controller and the drivetrain damping were made. As a result, results from FAST, BHawC, and the measurements show very good agreement in both the yaw and tilt directions. Particularly, FAST and BHawC results show excellent agreement in the mean as well as the standard deviations in the tilt direction, while the agreement between FAST and the experimental data is excellent in the yaw direction. As noted in the previous study,

the moments in nonrotating coordinates are computed using the rotor azimuth and the moments in the rotating frame, and



it therefore borrows from the uncertainties from the two quantities. Uncertainty quantification is outside of the scope of the current study but it is useful to consider while comparing results from the codes to the experimental data.

Figure 10 shows the tower-bottom side-side (perpendicular to the prevailing wind direction) bending moments. Another improvement to the FAST model mentioned earlier was the rotor mass imbalance that has been corrected in the model used

in this paper. Improvements due to this correction can be observed in the tower-bottom side-side bending moments shown in Figure 10.

Figure 11 shows the tower-top torsional moment. At lower wind speeds, the agreement between FAST with BeamDyn and the experimental data is very good. During at-rated and above-rated wind speeds, there seems to be a good agreement between the FAST with BeamDyn and the BHawC results, but there also shows a deviation from the data. It is noteworthy to mention

here that some uncertainty in the sensor calibration for this particular channel had been identified previously. Once again, these measurements are sensitive to the vertical shear as well as the yaw error of the turbine, along with the difficulty associated with experimental measurement of this QOI, making a comparison of the mean values inherently challenging. Overall, the comparison between the numerical predictions of this QOI using FAST and BHawC shows good agreement.

### 3.1.2   Discussion of results in the frequency domain

Figures 12 to 22 show the PSDs of the different QOIs in three different bins, at below-rated, rated, and above-rated mean hub-height inflow wind speeds. Each PSD plot represents the average of individual PSDs of the several 10 min data sets in the respective bin: for example, the PSD of a QOI at 8 m s$^{-1}$ represents the average from 103 PSDs corresponding to the 103 individual data sets, according to Table 1. Each figure shows six PSD plots: the plots to the left (subplots *a*, *c*, and *e*) show comparisons among FAST with BeamDyn, BHawC, and experimental measurements, and the plots to the right (subplots *b*,

*d*, and *f*) show comparisons between FAST with BeamDyn and FAST with ElastoDyn. The top two plots (subplots *a* and *b*) represent the 8 m s$^{-1}$ wind speed bin, which is below-rated operation. The two plots in the middle (subplots *c* and *d*) represent the 12 m s$^{-1}$ wind speed bin, which is approximately at-rated operation. The two plots to the bottom (subplots *e* and *f*) represent the 15 m s$^{-1}$ bin, which is above-rated operation. As highlighted before, the $x$-axes represent frequency and the values are hidden to protect Siemens data confidentiality. It has been highlighted in several previous studies that frequencies that are lower than

approximately 10 times the rotor rated speed (1P) contribute most to fatigue loads (up to 2 Hz for turbines with 1P = 0.2 Hz; see Sim et al. (2012); Bergami and Gaunaa (2014)), and so the range shown in the figures in this paper includes the frequencies most important for loads analysis. The peaks representing the frequencies with the highest energy in each figure have been highlighted and labeled as *A*, *B*, *C*, etc., and the comparisons among FAST, BHawC, and the measurements presented in these figures are discussed relative to these peaks in the following paragraphs.

Figures 12 and 13 represent the PSDs of the electrical power and the rotor speed, respectively. Peaks *A* and *C* in Figure 12 represent the rotor frequencies 1P and 3P, respectively. Note that this rotor operates at 16 rpm at rated power (1P $\simeq$ 0.267 Hz). Peak *B* seems to be the tower side-side mode that couples with the rotor speed as well as the shaft torque, and peak *D* is most likely the drivetrain torsional mode. Peaks *A*, *B*, and *C* match quite well among FAST, BHawC, and the data. While FAST results are close to the BHawC results or the measurements, the small difference between them may be due to the difference





in the drivetrain models used. FAST results show higher energy in between the labeled peaks, particularly at lower frequencies and higher wind speeds. The reason for it is not certain−it may be a numerical artifact, or due to the controller interface to FAST, but in any case it does not have a significant effect on either the dominant frequencies, or the mean and the standard deviation values, as can be seen from Figures 1 and 2.

Similarly, peaks $A$ and $D$ in Figure 13 represent the rotor frequencies 1P and 3P, respectively; peak $E$ the drivetrain torsional mode and peak $B$ the tower side-side mode−all of which match quite well among FAST, BHawC, and the measurements. FAST results show some energy at peak $C$ around the 2P frequency, which is not visible in the data or BHawC results. The exact reason for its presence in FAST and not the other channels is not certain, but considering the $y$-axis is a log scale, this peak has much less energy than the other dominant frequencies seen in the same plots.

PSDs of the rotor thrust force plotted in Figure 14 also show a good agreement between FAST and the data at the 1P and 3P frequencies denoted by peaks $A$ and $C$, respectively, as well as at peak $B$, which is the first tower fore-aft mode. Note that in the lower wind speed bin shown in Figure 14a, the rotor speed varies depending on the instantaneous wind speed, which is why several peaks are noticed in the region immediately below 1P.

The main difference between the two blade models used in the FAST simulations, ElastoDyn and BeamDyn, can most
clearly be observed in the blade-root bending moments. The PSD of the blade-root in-plane bending moments shown in Figure 17 shows two distinct features highlighting the improvements of using BeamDyn. Firstly, ElastoDyn simulations show the first blade-root in-plane bending moments at a higher value at peak $D$ rather than peak $C$ shown by BeamDyn. Secondly, the peak at $E$, which is most likely a coupled blade in-plane and torsional modes, are reproduced by BeamDyn simulations but not the model using ElastoDyn. Other peaks with the highest energies are peaks $A$ and $F$, which represent 1P and 2P, respectively, and
$B$, which is most likely the first blade edge-wise mode. FAST with BeamDyn shows good agreement with the data for these modal frequencies during operation. There is a difference between FAST and the data outside the peaks, but this difference is ignored for the current analysis because, as mentioned before, the $y$-axes used here are on a log scale and so these differences are very small compared to the energy at 1P.

The PSDs of the blade-root out-of-plane bending moments are shown in Figure 18. The peaks $A$, $B$, and $C$ denote 1P, 2P,
and 3P, respectively. Because the frequencies in the out-of-plane direction for a tilted rotor are primarily dominated by shear and gravity and because we also know that the rotor speeds match in all simulations, FAST (with ElastoDyn and BeamDyn), BHawC, and the measurements exhibit a similar behavior for this QOI.

The PSDs of the blade-tip deflections, shown in Figures 15 and 16, follow the trends observed in the blade-root bending moments. In Figure 15, the peaks $A$ and $B$ denote 1P and 2P, respectively; $C$ represents the first blade-wise mode; and $D$
represents the blade edge coupled with the drivetrain torsional mode. In Figure 16, the peaks $A$, $B$, and $C$ represent 1P, 2P, and 3P, respectively. The agreement among FAST with BeamDyn, BHawC, and the measurements is very good in both the in-plane as well as out-of-plane tip deflections. A comparison between ElastoDyn and BeamDyn results in Figures 4b, 4d, and 15d reveals significant improvement in BeamDyn over ElastoDyn in accurately capturing the frequencies of those modes that are coupled with the blade edge−namely peaks $C$ and $D$.





Figures 19 and 20 show the PSDs of the main-shaft bending moments in the yaw and the tilt directions, respectively. The most dominant frequency at peak *B* matches well with the experiments and the data in both yaw and tilt directions, which is most likely the first shaft-torsion mode. The peak denoted by *C*, which is most likely a blade-edge bending mode coupled to the shaft torsion, is visible and in agreement in the 8 m s$^{-1}$ case among FAST, BHawC, and the measurements, while at higher

5  wind speeds, the peak in the FAST data is not as conspicuous. Also, the 1P peak denoted by *A* is not as pronounced in the FAST simulations when compared to the data. These differences are most likely due to the combination of the filter used on the high-speed shaft in the FAST simulations to overcome the issues related to the controller, as mentioned earlier, and the limitations of the single-torsion-DOF drivetrain model used in FAST as opposed to a real drivetrain with multiple modes.

The PSDs of the tower-bottom side-side bending moments plotted in Figure 21 show a good match of the 1P denoted by

10  peak *A*, as well as the first tower side-side mode denoted by peak *B*. The PSD of the tower-top torsional moments in Figure 22 also shows good agreement among FAST, the data, and BHawC results at the most prominent peaks in *A* and *B*. Higher torsional modes, such as peak *C*, shown by BHawC and the measurements, are not modeled by ElastoDyn.





**Figure 1.** Electrical power.
**(a)** Individual 10 min averages from experimental data, BHawC, and FAST (BD).
**(b)** Binned averages of data in (a) are denoted by points connected by lines; the scatter above and below these data denote plus or minus one standard deviation, respectively, of the individual 10 min data sets in that bin.
**(c)** Individual 10 min averages from FAST (BD) and FAST (ED).
**(d)** Binned averages of data in (c) are denoted by the points connected by lines; the scatter above and below these data denote plus or minus one standard deviation, respectively, of the individual 10 min data sets in that bin.



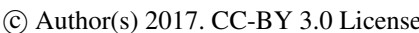

**Figure 2.** Rotor speed.
**(a)** Individual 10 min averages from experimental data, BHawC, and FAST (BD).
**(b)** Binned averages of data in (a) are denoted by points connected by lines; the scatter above and below these data denote plus or minus one standard deviation, respectively, of the individual 10 min data sets in that bin.
**(c)** Individual 10 min averages from FAST (BD) and FAST (ED).
**(d)** Binned averages of data in (c) are denoted by the points connected by lines; the scatter above and below these data denote plus or minus one standard deviation, respectively, of the individual 10 min data sets in that bin.



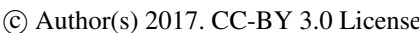



**Figure 3.** Rotor thrust force.
**(a)** Individual 10 min averages from experimental data, BHawC, and FAST (BD).
**(b)** Binned averages of data in (a) are denoted by points connected by lines; the scatter above and below these data denote plus or minus one standard deviation, respectively, of the individual 10 min data sets in that bin.
**(c)** Individual 10 min averages from FAST (BD) and FAST (ED).
**(d)** Binned averages of data in (c) are denoted by the points connected by lines; the scatter above and below these data denote plus or minus one standard deviation, respectively, of the individual 10 min data sets in that bin.





**Figure 4.** Blade in-plane tip deflections.
**(a)** Individual 10 min averages from experimental data, BHawC, and FAST (BD).
**(b)** Binned averages of data in (a) are denoted by points connected by lines; the scatter above and below these data denote plus or minus one standard deviation, respectively, of the individual 10 min data sets in that bin.
**(c)** Individual 10 min averages from FAST (BD) and FAST (ED).
**(d)** Binned averages of data in (c) are denoted by the points connected by lines; the scatter above and below these data denote plus or minus one standard deviation, respectively, of the individual 10 min data sets in that bin.





**Figure 5.** Blade out-of-plane tip deflections.
**(a)** Individual 10 min averages from experimental data, BHawC, and FAST (BD).
**(b)** Binned averages of data in (a) are denoted by points connected by lines; the scatter above and below these data denote plus or minus one standard deviation, respectively, of the individual 10 min data sets in that bin.
**(c)** Individual 10 min averages from FAST (BD) and FAST (ED).
**(d)** Binned averages of data in (c) are denoted by the points connected by lines; the scatter above and below these data denote plus or minus one standard deviation, respectively, of the individual 10 min data sets in that bin.

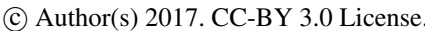



**Figure 6.** Blade-root in-plane bending moments.
**(a)** Individual 10 min averages from experimental data, BHawC, and FAST (BD).
**(b)** Binned averages of data in (a) are denoted by points connected by lines; the scatter above and below these data denote plus or minus one standard deviation, respectively, of the individual 10 min data sets in that bin.
**(c)** Individual 10 min averages from FAST (BD) and FAST (ED).
**(d)** Binned averages of data in (c) are denoted by the points connected by lines; the scatter above and below these data denote plus or minus one standard deviation, respectively, of the individual 10 min data sets in that bin.





**Figure 7.** Blade-root out-of-rotor-plane bending moments.
**(a)** Individual 10 min averages from experimental data, BHawC, and FAST (BD).
**(b)** Binned averages of data in (a) are denoted by points connected by lines; the scatter above and below these data denote plus or minus one standard deviation, respectively, of the individual 10 min data sets in that bin.
**(c)** Individual 10 min averages from FAST (BD) and FAST (ED).
**(d)** Binned averages of data in (c) are denoted by the points connected by lines; the scatter above and below these data denote plus or minus one standard deviation, respectively, of the individual 10 min data sets in that bin.



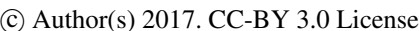

**Figure 8.** Main-shaft bending moment−yaw (nonrotating coordinate system).
**(a)** Individual 10 min averages from experimental data, BHawC, and FAST (BD).
**(b)** Binned averages of data in (a) are denoted by points connected by lines; the scatter above and below these data denote plus or minus one standard deviation, respectively, of the individual 10 min data sets in that bin.
**(c)** Individual 10 min averages from FAST (BD) and FAST (ED).
**(d)** Binned averages of data in (c) are denoted by the points connected by lines; the scatter above and below these data denote plus or minus one standard deviation, respectively, of the individual 10 min data sets in that bin.



**Figure 9.** Main-shaft bending moment−tilt (nonrotating coordinate system).
**(a)** Individual 10 min averages from experimental data, BHawC, and FAST (BD).
**(b)** Binned averages of data in (a) are denoted by points connected by lines; the scatter above and below these data denote plus or minus one standard deviation, respectively, of the individual 10 min data sets in that bin.
**(c)** Individual 10 min averages from FAST (BD) and FAST (ED).
**(d)** Binned averages of data in (c) are denoted by the points connected by lines; the scatter above and below these data denote plus or minus one standard deviation, respectively, of the individual 10 min data sets in that bin.



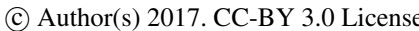


**Figure 10.** Tower-bottom side-side bending moments.
**(a)** Individual 10 min averages from experimental data, BHawC, and FAST (BD).
**(b)** Binned averages of data in (a) are denoted by points connected by lines; the scatter above and below these data denote plus or minus one standard deviation, respectively, of the individual 10 min data sets in that bin.
**(c)** Individual 10 min averages from FAST (BD) and FAST (ED).
**(d)** Binned averages of data in (c) are denoted by the points connected by lines; the scatter above and below these data denote plus or minus one standard deviation, respectively, of the individual 10 min data sets in that bin.





**Figure 11.** Tower-top torsion moment.
**(a)** Individual 10 min averages from experimental data, BHawC, and FAST (BD).
**(b)** Binned averages of data in (a) are denoted by points connected by lines; the scatter above and below these data denote plus or minus one standard deviation, respectively, of the individual 10 min data sets in that bin.
**(c)** Individual 10 min averages from FAST (BD) and FAST (ED).
**(d)** Binned averages of data in (c) are denoted by the points connected by lines; the scatter above and below these data denote plus or minus one standard deviation, respectively, of the individual 10 min data sets in that bin.







**Figure 12.** PSDs of the electrical power. The three sets of graphs represent the ensemble-averaged PSDs from the inflow velocity bins at 8 m s$^{-1}$ (top), 12 m s$^{-1}$ (center), and 15 m s$^{-1}$ (bottom).





**Figure 13.** PSDs of the rotor speed. The three sets of graphs represent the ensemble-averaged PSDs from the inflow velocity bins at 8 m s$^{-1}$ (top), 12 m s$^{-1}$ (center), and 15 m s$^{-1}$ (bottom).





**Figure 14.** PSDs of the rotor thrust force. The three sets of graphs represent the ensemble-averaged PSDs from the inflow velocity bins at 8 m s$^{-1}$ (top), 12 m s$^{-1}$ (center), and 15 m s$^{-1}$ (bottom).





**Figure 15.** PSDs of the in-plane blade-tip deflection. The three sets of graphs represent the ensemble-averaged PSDs from the inflow velocity bins at 8 m s$^{-1}$ (top), 12 m s$^{-1}$ (center), and 15 m s$^{-1}$ (bottom).



**Figure 16.** PSDs of the out-of-plane blade-tip deflection. The three sets of graphs represent the ensemble-averaged PSDs from the inflow velocity bins at 8 m s⁻¹ (top), 12 m s⁻¹ (center), and 15 m s⁻¹ (bottom).






**Figure 17.** PSDs of the blade-root in-plane bending moments. The three sets of graphs represent the ensemble-averaged PSDs from the inflow velocity bins at 8 m s$^{-1}$ (top), 12 m s$^{-1}$ (center), and 15 m s$^{-1}$ (bottom).




**Figure 18.** PSDs of the blade-root out-of-rotor-plane bending moments. The three sets of graphs represent the ensemble-averaged PSDs from the inflow velocity bins at 8 m s$^{-1}$ (top), 12 m s$^{-1}$ (center), and 15 m s$^{-1}$ (bottom).







**Figure 19.** PSDs of the main-shaft bending moment−yaw (nonrotating coordinate system). The three sets of graphs represent the ensemble-averaged PSDs from the inflow velocity bins at 8 m s⁻¹ (top), 12 m s⁻¹ (center), and 15 m s⁻¹ (bottom).







**Figure 20.** PSDs of the main-shaft bending moment−tilt (nonrotating coordinate system). The three sets of graphs represent the ensemble-averaged PSDs from the inflow velocity bins at 8 m s$^{-1}$ (top), 12 m s$^{-1}$ (center), and 15 m s$^{-1}$ (bottom).







**Figure 21.** PSDs of the tower-bottom side-side bending moments. The three sets of graphs represent the ensemble-averaged PSDs from the inflow velocity bins at 8 m s$^{-1}$ (top), 12 m s$^{-1}$ (center), and 15 m s$^{-1}$ (bottom).







**Figure 22.** PSDs of the tower-top torsion moment. The three sets of graphs represent the ensemble-averaged PSDs from the inflow velocity bins at 8 m s$^{-1}$ (top), 12 m s$^{-1}$ (center), and 15 m s$^{-1}$ (bottom).





## 4 Conclusion

In this paper, the latest results from FAST v8 with the new BeamDyn and AeroDyn modules have been presented. The release of BeamDyn and the AeroDyn overhaul within FAST v8 opens up new possibilities in modeling and designing advanced aeroelastically tailored blades. Following the IEC-61400-13 standard as a guideline, a comparison has been presented between 5 FAST with and without its latest improvements, Siemens' in-house code BHawC, and the experimental data that were acquired from a series of field-test measurements from the Siemens 2.3 MW wind turbine at the NWTC.

Tables 2 and 3 show the tools FAST with BeamDyn, FAST with ElastoDyn, and BHawC, ranked according to how well their results compare qualitatively with the experimental measurements−the lower the rank, the better the agreement with the measured data. Note that this ranking is purely qualitative, and does not take into account any measurement uncertainties in 10 the experimental data. Results reveal significant improvements in the modeling capabilities of the latest version of FAST over its previous capabilities in all of the QOIs investigated in this paper. From Table 2, FAST with BeamDyn shows excellent agreement with the measurements in the time domain in general. This agreement is significantly closer to the measurements than ElastoDyn simulations, and on par with and in some cases marginally closer to the measurements than BHawC. From Table 3, FAST with BeamDyn is shown to consistently deliver more accurate predictions compared to ElastoDyn. The FAST 15 simulations with BeamDyn do not compare as well with the experimental data as BHawC, but this is most likely because of the difference in the controller and the drivetrain models.

Based on the presented results overall, FAST v8 has been sufficiently validated against field measurements, along with a code-to-code verification with BHawC. FAST v8 has been demonstrated to be valid for aeroelastic loads analyses of wind turbines, even with blades with significant aeroelastic tailoring.

20 The results did not include uncertainties that are inherent to the experimental data acquisition process as well as the modeling and simulation process. For example, there may be errors due to the differences in the controller used in the simulation compared to the actual machine; there may be uncertainties associated with the structural properties of the blades, tower, drivetrain, etc.; there may be uncertainties due to the limitations in the information available about the inflow wind and shear conditions measured that were used to recreate the turbulence boxes; and there may be a measurement error that has not been quantified 25 in this paper. Analysis of all such uncertainties and their impact on the results requires a detailed, separate work, which will be a part of future work. In addition, validation in conditions with elevated yaw errors and wind speed variations, both associated with extreme conditions, correspond to disparate physics and numerics and require special considerations, which will also be a part of future work.

*Acknowledgements.* This work is the outcome of a collaborative research and development agreement between NREL and Siemens Wind 30 Power, CRADA No. CRD-08-303. The submitted manuscript has been offered by employees of the Alliance for Sustainable Energy, LLC (Alliance), a contractor of the U.S. Government under Contract No. DE-AC36-08GO28308. The U.S. Government retains and the publisher, by accepting the article for publication, acknowledges that the U.S. Government retains a nonexclusive, paid-up, irrevocable, worldwide license to publish or reproduce the published form of this work, or allow others to do so, for U.S. Government purposes.



| QOI (Time domain) | FAST (BD) | FAST (ED) | BHawC |
|---|---|---|---|
| Elec. power | - | - | - |
| Rotor speed | 2 | 2 | 1 |
| Rotor thrust | 1 | 3 | 1 |
| Blade in-plane tip deflection | 1 | 3 | 2 |
| Blade out-of-plane tip deflection | 1 | 3 | 2 |
| Blade-root in-plane bending moment | - | - | - |
| Blade-root out-of-rotor-plane bending moment | 2 | 3 | 1 |
| Main-shaft bending moment - yaw | 1 | 3 | 1 |
| Main-shaft bending moment - tilt | 1 | 2 | 3 |
| Tower-bottom side-side bending moment | - | - | - |
| Tower-top torsion moment | 1 | 3 | 1 |

**Table 2.** The tools FAST (BD), FAST (ED), and BHawC ranked according to how well their results compare to the experimental measurements in the time domain for each QOI. No value indicates that no discernible difference could be seen among the different tools for that QOI.

| QOI (Freq. domain) | FAST (BD) | FAST (ED) | BHawC |
|---|---|---|---|
| Elec. power | 2 | 3 | 1 |
| Rotor speed | 2 | 3 | 1 |
| Rotor thrust | 2 | 2 | 1 |
| Blade-root in-plane bending moment | 1 | 3 | 1 |
| Blade-root out-of-rotor-plane bending moment | - | - | - |
| Blade in-plane tip deflection | 1 | 3 | 1 |
| Blade out-of-plane tip deflection | - | - | - |
| Main-shaft bending moment - yaw | 2 | 2 | 1 |
| Main-shaft bending moment - tilt | 2 | 2 | 1 |
| Tower-bottom side-side bending moment | 2 | 3 | 1 |
| Tower-top torsion moment | 2 | 2 | 1 |

**Table 3.** The tools FAST (BD), FAST (ED), and BHawC ranked according to how well their results compare to the experimental measurements in the frequency domain for each QOI. No value indicates that no discernible difference could be seen among the different tools for that QOI.





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
