# Peer review of "A Validation and Code-to-Code Verification of FAST for a Megawatt-Scale Wind Turbine with Aeroelastically Tailored Blades"

_Wind Energy Science, 2016_

## Referee Comment (RC1) · Anonymous Referee #1 · 25 Apr 2017

Dear Editor,

overall the paper is quite interesting and addresses a relevant subject for wind industry, which is aeroelastic simulation and the validation of the simulation tools. The availability of experimental data for the discussion is a big plus since such data are normally quite rare, especially for large machines.

The only doubt i have is about Siemens tool role in this work. It is of course interesting to see a benchmark between a proprietary tool and FAST, but by definition of proprietary tool, details about the implementation and the modelling are not really presented in detail. The tool itself is of course not available. From Siemens point of view, it is understandable the PR effect of showing that their tool is quite in agreement with experiments, but what is the main scientific value of including Siemens tool in this work ? Said in other words, the data about Siemens simulations are nice-to-see but they don't add value to the work because the tool details are not presented/discussed. My recommendation is either to provide details for Siemens tool which are beneficial for the reader in order to understand the results, or remove the data from the graphs

———————————————

---

## Referee Comment (RC2) · Anonymous Referee #2 · 12 Jun 2017

Dear authors and editor,

Overall the article is very interesting and well-written. The analysis of the different channels is quite detailed and the authors explain most of the observed deviations between simulations and experiments.

I don't quite understand the distinction between tables 2 and 3. How can the agreement of the different codes with experiments be better or worse in the time domain and frequency domain? Since the analysis is based on the same results, the agreement should be the same, no matter if the results are analysed in time- or frequency domain. I would suggest to combine Tables 2 and 3 in one Table showing the overall agreement

or elaborate more on the different criteria applied in Tables 2 and 3.

To my opinion, a revised version of the article should include:

1) An explanation of why the rated rotor speed in the FAST simulations deviates slightly from the experimental data and the BHawC results.

2) If possible some rough details on the BHawC aerodynamic model: I guess it is BEM based and employs some kind of dynamic stall / unsteady airfoil aerodynamics model. As an example, it is stated about the structural modeling in BHawC in the article: 'In contrast to the nonlinear finite-element implementation of BeamDyn, the structural model of BHawC employs a co-rotational beam formulation, which is a combined multibody and linear finite-element representation allowing for geometric nonlinearities through a series of multiple bodies, each composed of a linear finite element. ' This gives some indication of the capabilities without giving away too much detail. A similar description of the aerodynamics part would be beneficial.

3) A discussion on the tower side-side bending moments. Quite some difference in the standard deviations are visible in Figure 10 at wind speeds between 15 and 19 m/s. FAST predicts much higher standard deviations, which is also visible in the PSD in Figure 21. In Table 2 this difference is not mentioned.

Please find additional comments in the annotated paper that is attached to this report.

Please also note the supplement to this comment:
http://www.wind-energ-sci-discuss.net/wes-2016-42/wes-2016-42-RC2-supplement.pdf

**Supplement:**

[revised manuscript text omitted]

---

## Author Comment (AC1) · 11 Jul 2017

Dear Reviewer 1,

Thank you for your time and valuable comments on our manuscript. Here are our responses to your comments.

Comment: ". . . my recommendation is either to provide details for Siemens tool which are beneficial for the reader in order to understand the results, or remove the data from the graphs."

Response: We will add the following information that gives more details on BHawC,

along with the relevant citation that contains more details about BHawC:

Description of BHawC:

BHawC is an aeroelastic simulation tool used to study the dynamic response of wind turbines. The model consists of substructures for foundation, tower, nacelle, drivetrain, gearbox, hub, and blades. The structure is modelled primarily with finite beam elements and the aerodynamics is modelled using blade element momentum theory. The code is coupled to a controller identical to that on the real turbine.

The structural model of BHawC employs a co-rotational beam formulation, which is a combined multibody and linear finite-element representation allowing for geometric nonlinearities through a series of multiple bodies, each composed of a linear finite element. The BHawC model of the SWT-2.3-108 blade used in the current study was initially curved in space and discretized into 16 linear elements. In other parts of the turbine where bearings are present, special elements are introduced and the drivetrain consists purely of torsional elements. The aerodynamic force in BHawC is calculated at a given number of points on the blades, in this case 63, positioned independently of the structural nodes. Blade element momentum theory is applied to determine the tangential and axial induced velocities at these aerodynamic calculation points, and Prandtl's tip loss correction as well as a correction for thrust at high induction values are implemented. The blade-element implementation in BHawC also allows for unsteady and skewed inflow. The aerodynamic force is based on 3D-corrected coefficients for stationary airfoil data, and a Beddoes-Leishman type model for unsteady/dynamic events. In addition, BHawC contains a model for tower shadow, and it also calculates the aerodynamic forces on the nacelle and tower. For further details on BHawC , see [1].

[1] Skjoldan, Peter Fisker. "Aeroelastic modal dynamics of wind turbines including anisotropic effects." PhD Thesis, Risø-PhD-66(EN), March 2011. http://orbit.dtu.dk/fedora/objects/orbit:85866/datastreams/file_5509069/content

Sincerely, Srinivas Guntur.

---

## Author Comment (AC2) · 11 Jul 2017

Dear Reviewer 2,

Thank you for your time and valuable comments on our manuscript. Here are our responses to your comments.

Comment: "I don't quite understand the distinction between tables 2 and 3. How can the agreement of the different codes with experiments be better or worse in the time domain and frequency domain " Response: The intention behind making two tables was to be able to distinguish between analysis in the time domain (Table 2), which

ranks the results in the mean and the standard deviation of the time series of various quantities, and analysis in the frequency domain (Table 3), which ranks the turbine operating modes. The title of Table 2 will be changed to

"The tools FAST (BD), FAST (ED), and BHawC ranked according to how well their results compare to the experimental measurements in the mean and standard deviation for each QOI. . . ",

and that of Table 3 to

"The tools FAST (BD), FAST (ED), and BHawC ranked according to how well their PSD results compare to the experimental measurements . . .",

in order to avoid any confusion.

Comment: "An explanation of why the rated rotor speed in the FAST simulations deviates slightly from the experimental data and the BHawC results." Response: This slight difference is most likely the result of the controller (a DLL in a black-box form) that was obtained from Siemens to carry out the FAST simulations. It is assumed (with reasonable confidence) that this difference doesn't have a significant effect on the conclusions of this paper.

Comment: "If possible some rough details on the BHawC aerodynamic model" Response: A description of BHawC and a reference have been added (see below).

Comment: A discussion on the tower side-side bending moments (+ comments on Figure 10). Response: The emphasis in the paper is to identify the correct frequency of this mode (which is a function of the dynamic system modelled in FAST, which includes the combined dynamics of blade elements, hub, nacelle, generator, and the tower), and whether or not the frequency is being damped out is most likely related to the controller. The difference seen in the tower side-side moments seem to be related to the latter. Since the paper is focussed on the modelling ability of FAST (along with our inability to change the controller settings), we focussed on it being the correct freq. This has

been highlighted in text thus: "Plots c and e in Figure 21 show a difference in the peak amplitude at peak B. This reflects the difference in the standard deviation seen in Figure 10 between FAST and the measurements. As seen, the frequency of this mode is accurately captured by FAST. This mode seems to be less-damped than in BHawC or the measurements, which may be related to the tuning of the DLL controller."

Comment: What is the difference between aeroelastic tailoring and bend-twist coupling? Response: The two terms imply a similar technology, although bend-twist-coupling is a subset of aeroelastic tailoring. This sentence has been changed to the following to avoid confusion: "Additionally, these blades are flexible and aeroelastically tailored, incorporating bend-twist coupling."

Comment: Suggestion to change 'simplicity' to brevity' Response: Done

Comment: Page 9., Line 21. "due to weight" Response: Added.

Comment: Is it possible to include pitch angles also? Response: Unfortunately, pitch-power and pitch-speed curves are deemed confidential and cannot be published.

Comment: Is the main difference here (Figure 3d) due to torsion? Response: Blade torsion was not investigated in this paper (due to the lack of blade-twist data), but that is very much possible. However, we see significant differences between ElastoDyn and BeamDyn in terms of the tip deflections (e.g., Figure 4d), and so it may be caused by blade bending and also torsion.

Comment: Change "all" to "most of the QOIs" in Conclusions Response: Done.

– Description of BHawC:

BHawC is an aeroelastic simulation tool used to study the dynamic response of wind turbines. The model consists of substructures for foundation, tower, nacelle, drivetrain, gearbox, hub, and blades. The structure is modelled primarily with finite beam elements and the aerodynamics is modelled using blade element momentum theory. The code is coupled to a controller identical to that on the real turbine.

The structural model of BHawC employs a co-rotational beam formulation, which is a combined multibody and linear finite-element representation allowing for geometric nonlinearities through a series of multiple bodies, each composed of a linear finite element. The BHawC model of the SWT-2.3-108 blade used in the current study was initially curved in space and discretized into 16 linear elements. In other parts of the turbine where bearings are present, special elements are introduced and the drivetrain consists purely of torsional elements.

The aerodynamic force in BHawC is calculated at a given number of points on the blades, in this case 63, positioned independently of the structural nodes. Blade element momentum theory is applied to determine the tangential and axial induced velocities at these aerodynamic calculation points, and Prandtl's tip loss correction as well as a correction for thrust at high induction values are implemented. The blade-element implementation in BHawC also allows for unsteady and skewed inflow. The aerodynamic force is based on 3D-corrected coefficients for stationary airfoil data, and a Beddoes-Leishman type model for unsteady/dynamic events. In addition, BHawC contains a model for tower shadow, and it also calculates the aerodynamic forces on the nacelle and tower. For further details on BHawC , see [1].

[1] Skjoldan, Peter Fisker. "Aeroelastic modal dynamics of wind turbines including anisotropic effects." PhD Thesis, Risø-PhD-66(EN), March 2011. http://orbit.dtu.dk/fedora/objects/orbit:85866/datastreams/file_5509069/content

–

Sincerely, Srinivas Guntur.

---

## Author Response (AR1)

Dear reviewers,

Thank you for your time and valuable comments on our manuscript. Here are the changes we incorporated into our revised paper, in accordance with the received comments.

1. The titles of Table 2 have been be changed to

   *"The tools FAST (BD), FAST (ED), and BHawC ranked according to how well their results compare to the experimental measurements in the mean and standard deviation for each QOI… "*,

   and Table 3 to

   *"The tools FAST (BD), FAST (ED), and BHawC ranked according to how well their PSD results compare to the experimental measurements …"*.

2. Explanation of results referring to Figure 21 has been changed to:
   *"Plots c and e in Figure 21 show a difference in the peak amplitude at peak B. This reflects the difference in the standard deviation seen in Figure 10 between FAST and the measurements. As seen, the frequency of this mode is accurately captured by FAST. This mode seems to be less-damped than in BHawC or the measurements, which may be related to the tuning of the DLL controller."*

3. Modification to page 3, line 6:
   *"Additionally, these blades are flexible and aeroelastically tailored, incorporating bend-twist coupling."*

4. Page 5, line 2: Changed the word simplicity to brevity.

5. Page 9., Line 35. Added "due to weight" in the explanation of why the blade-root in-plane moments are so heavily sinusoidal.

6. Changed "all" to "most of the QOIs" in Conclusions

7. Added Description of BHawC, section 2.2.

Sincerely,

Srinivas Guntur.